# Factors associated with antenatal depression in the Kingdom of Jordan during the COVID-19 pandemic

**Sanaa Abujilban**[1], **Lina Mrayan**[1], **Salwa Al-Obeisat**[2], **Mu'ath Tanash**[3], **Marlene Sinclair**[4], **W. George Kernohan**[4]*

1 Department of Maternal, Child and Family Health Nursing, Faculty of Nursing, The Hashemite University, Zarqa, Jordan, 2 Department of Maternal and Child Nursing, Faculty of Nursing, Jordan University of Science and Technology, Irbid, Jordan, 3 Department of Adult Health Nursing, Faculty of Nursing, The Hashemite University, Zarqa, Jordan, 4 Institute of Nursing and Health Research, Ulster University, Newtownabbey, County Antrim, Northern Ireland, United Kingdom

* wg.kernohan@ulster.ac.uk

## Abstract

Fear of infection and measures taken to mitigate infection, such as social distancing, lockdown and isolation can lead to anxiety and depression across the life course, but especially in pregnancy. We set out to identify the prevalence of depression in pregnancy, using Edinburgh Postnatal Depression Scale (EPDS) during national quarantine and to examine women's knowledge, attitude, and practices (KAP) in regard to potential COVID-19-related depression. Following ethical approval, an observational design, with an online questionnaire and snowball sampling was used to recruit 546 pregnant women (231 primi and 315 multiparous) in Jordan via common social media platforms (facebook, WhatsApp). Over one third (36.7%) reported depressive symptoms. There were significantly lower depression scores among pregnant women who exhibited more knowledge about COVID-19 (*in high [vs low] knowledge groups, mean EPDS = 10.8 [vs 12.2]; p = 0.007*). Depression scores were not significantly associated with attitude nor with practice. This suggests that enhanced knowledge levels may protect pregnant women against depression. Our findings contribute to understanding of the experience of pregnant women in developing countries during the COVID-19 pandemic. Healthcare Professionals should provide health education to all pregnant women and timely services to pregnant women with depressive symptoms. This may lead to the prevention of serious symptoms and reduce negative consequences on the next generation, not only in Jordan, but worldwide.

**Data Availability Statement:** All relevant data is available here: Abujilban, S., Mrayan, L., Al-Obeisat, S., Tanash, M., Marlene Sinclair, M., Kernohan, W.

## Introduction

Studies on antenatal mental health during the COVID- 19 pandemic are lacking, particularly in developing countries. It is important to address the sharp rise of depressive symptoms during the pandemic and to identify any association with women's knowledge, attitude, and practices in response to both COVID-19 and measures taken to control it.

G. Factors influencing levels of antenatal depression in the Kingdom of Jordan during the COVID-19 pandemic (dataset). 2021. Ulster University Institutional Repository (PURE). http://doi.org/10.21251/d886efdf-d688-4ca3-8151-7e5f5cd5d8be.

**Funding:** The authors received no specific funding for this work.

**Competing interests:** The authors have declared that no competing interests exist.

Compared to men, around double the number of women experience depression, and this difference has been variously attributed to their female sex hormones, the burdens of pregnancy, childbirth, and motherhood [1]. Antenatal depression is a common complication during pregnancy [2], although many depressed pregnant women are unaware of it [3]. According to the World Health Organization (WHO), 10% of pregnant women around the world and 15.6% in developing countries experience a mental health disorder, primarily depression [4]. In one review of the burden of antenatal depression Dadi, Miller [5], it was reported that the global prevalence of antenatal depression ranges from 15% to 65%. Moreover, the prevalence has been shown to increase as pregnancy progresses: one study reported 7.4% in the first trimester and 12.0%–12.8% in the second and third trimesters [2]. One report from Jordan suggests that pregnant women may have double the rate of antenatal depression (57%), compared to their counterparts in developed countries [6].

Risk factors for antenatal depression include poor partner support, intimate partner violence, low income, younger age [7], low level of education, higher parity [6], unplanned pregnancy [8], single marital status, recent stressful life events, and difficult childhood [9].

Antenatal depression affects the health of both mother and baby. In the baby, it increases the risk for low birth weight [10], child psychopathology [11], sleep problems [12], inappropriate attachment [13], respiratory and gastrointestinal infections [14], and violent behavior in adolescence [15]. Depressed mothers tend to have more thoughts of harming themselves [16] and more preterm births [10]. Women with antenatal depression are more likely to have postnatal depression [17].

On 31st December 2019, China reported to WHO that a pneumonia of unknown cause had been detected in the city of Wuhan. On January 30th, 2020, WHO declared that the outbreak was a *Public Health Emergency of International Concern* and on 11th February 2020, WHO named a new coronavirus disease: COVID-19 [18]. In addition to concern over the disease itself, researchers have reported increasing worries, fear and apprehension about contracting the COVID-19 infection among the public, leading to mental health problems [19].

Worldwide, little is known about the effect of COVID-19 on women's psychological status and pregnancy outcomes. However, it has been suggested that some pregnant women have negative feelings such as stress, anxiety, fear, or uncertainty because of the COVID-19 pandemic [20], which could trigger depressive symptoms. In such an emergency situation, it is likely that women would experience more stress as a result of potential economic hardship and job loss, as well as an increase in the amount of caregiving because of school closures [21]. In Jordan, only two cases of COVID-19 in pregnant women had been registered (by 1 May 2020). Both underwent a successful caesarean section, with good birth outcomes and no vertical transmission to the babies [22]. From the middle of March 2020 (at the time of data collection) till the end of April 2020, the government of Jordan has imposed strict quarantine restrictions with most people remaining under curfew in their homes. At the time of data collection, strict quarantine was applied in Jordan (for the whole population). Cases were rare (the total number of COVID-19 cases was 453 with 8 deaths [23] and were isolated in hospitals with no family or other visitors permitted. The current situation of COVID-19 in Jordan (until October 6, 2021) is 828,572 cases with 10,762 deaths [23].

In an extensive review of previous studies on COVID-19, we found that COVID-19 is affecting the mental health of pregnant women to a variable extent [24]. One Egyptian study found that 44.2% of pregnant women were experiencing depressive symptoms during the pandemic [24]. Similarly a study from China reported that 45.9% of pregnant women were having depressive symptoms during the COVID-19 outbreak [25]. In contrast, research in Greece found that the pandemic increased anxiety of pregnant women but not depression levels, which remain at 13.5% [26]. All of these studies used EPDS to measure depressive symptoms.

Although the situation in Jordan has not yet been fully explored, the government has disseminated public health educational messages on the prevention of COVID-19 through radio, television, and social media channels. If people have accurate knowledge and a positive attitude, it has been suggested that they will practice appropriate control measures [27]. Furthermore, accurate health information and appropriate precautionary measures are associated with lower levels of depression, anxiety, and stress, and thereby lower the psychological impact of the outbreak upon the general population [28].

Recent studies on pregnancy and COVID-19 have focused on the psychological effect of the disease. Poor psychological status could result in negative consequences for both mother and baby that could exceed those of the pandemic itself [29] and could have a lifelong effect on women's overall wellbeing [30]. Moreover, very little is known in the Jordanian context about the Knowledge, Attitude, and Practices (KAP) toward the coronavirus and its association with depressive symptoms among pregnant women. A clear understanding of the mental health of pregnant women during the pandemic is important for their long-term health [30]. Therefore, the aims of the current study were to identify the prevalence of antenatal depression and to explore possible associations between KAP toward COVID-19 and socio-demographics with antenatal depression symptoms among Jordanian pregnant women.

An understanding of the association of KAP toward COVID-19 with antenatal depression could assist policymakers to develop actionable policies and help nurses and midwives to provide timely services to the affected pregnant women [31], including attention and recovery programs [32]. Also, having an insight into the mental health response of pregnant women toward COVID-19 might help healthcare providers and communities to prepare a better response in such emergency situations [33].

The current study aimed to answer the following questions: What is the prevalence of antenatal depression during quarantine? Do KAP toward COVID-19 and socio-demographics have any association with antenatal depression symptoms?

## Materials and methods

### Design

A cross-sectional, correlational design was used for this study because the non-manipulative nature of the independent variables suited the study purpose.

### Sample

The target population was all pregnant women in Jordan, who met the following inclusion criteria: healthy pregnant women, who were from different reproductive age groups and social classes, who could read and write Arabic, who were willing to participate, and who could access and complete the online survey, following contact made through social networks (such as *Facebook* and *WhatsApp*, *see* S1 File). We excluded non-pregnant and non-Jordanian women. Snowballing sampling was used, where early participants passed on details of the study to other participants within their networks [34]. This was introduced as antenatal care ceased during the pandemic: only emergency cases were seen in hospital. The required sample size for the study was 500 based on Thorndike's rule-of-thumb: 20 participants for each variable [35].

### Instruments

Women completed the online Google Forms questionnaire (open survey) in three parts (see S2 and S3 Files). The usability and technical functionality of the electronic questionnaire had

been tested by piloting the questionnaire. The Checklist for Reporting Results of Internet E-Surveys (see S1 Checklist) [36] was followed with some limitations (see the Limitation section). The three parts are:

**Part I: Socio-Demographical, Obstetrical, and Gynecological History (DOGH).** A questionnaire was developed specifically for this study and eight experts in the field of maternity care, validated the DOGH for relevancy, non-ambiguity, simplicity and clarity. The content validity index was acceptable (0.83). Nominal, ordinal, interval, and ratio measurements were used.

**Part II: Assessment of depression.** The EPDS was used to record antenatal depression symptoms [37]. It is a 10-question, self-administered screening tool, where each item is scored on a four-point Likert-type scale range from 0 (not at all) to 3 (most of the time): a total score of 13 or above indicating the presence of antenatal depression symptoms [16]. The EPDS has been found to be valid and reliable for the measurement of depressive symptoms among Jordanian pregnant women ($\alpha = 0.79$) [38]. In the current study, the EPDS was found to have a Cronbach's alpha of 0.85, indicating high reliability [34]. As above, three experts in the field of maternity care, validated the items for relevancy, ambiguity, simplicity and clarity.

**Part III: Knowledge, Attitude, and Practices (KAP).** The KAP questionnaire used, with permission, was developed by Zhong, Luo [27]. Aspects of COVID-19 knowledge (clinical presentations, transmission routes, prevention, and control) were recorded by 12 items: each scored one point if answered correctly and zero for false or 'do not know'; a higher score indicating better knowledge. The internal consistency in the original study was acceptable ($\alpha = 0.71$). The English-language version of the scale was translated into Arabic and then back-translated into English to ensure accuracy. Attitudes and practices were measured by two items each. For the purpose of the current study another 25 knowledge items (total 37 items), 6 attitude items (total become 8 items), and 11 practice items (total become 13 items) were added. The extra items were based on The American College of Obstetricians and Gynecologists' "Coronavirus (COVID-19), Pregnancy, and Breastfeeding: A Message for Patients" report [20] which recommended specific information about COVID-19 for pregnant women. The additional items were validated by three PhD-qualified experts in maternal care, who agreed on the relevancy, clarity, and appropriateness of the questions for Jordanian culture. A copy of the complete questionnaire is available on request (Arabic or English).

## Data collection

Data were collected from pregnant women by using a voluntary online questionnaire (Google Forms) with no incentives. A link was sent to online groups set up for pregnancy and pregnant women, where pregnant women share their experiences with pregnancy with each others. The first page comprised an information sheet, explaining the women's rights, study risk and benefits, time needed to fill in the questionnaire, researcher contact information and consent to proceed to the questionnaire. The online form was composed of 15 sections. Each section was on a separate page with a back button to previous page. Adaptive questions were used as necessary to reduce number and complexity of the questions. All items were mandotory to ensure the completeness of the questionnaire, where some items have "not applicable" option. It was estimated to take approximately 10 minutes to complete the questionnaire. Data were collected during the full quarantine in March and April 2020.

## Analysis

Responses were entered automatically into a spreadsheet (MS-Excel), coded and then imported for analysis (IBM-SPSS, version 24). Descriptive statistics (frequencies, percentages,

means and standard deviations) were used to describe the characteristics of the participants and the main variables. T-tests were used to explore the association of KAP and socio-demographics with EPDS with p ≤ 0.05 taken as significant. Preliminary analyses ensured no violations of the assumptions of the tests [39]. The total sample size for the study was 546, more than sufficient to address the study purposes. This included 231 primiparous and 315 multiparous women.

## Ethical considerations

The Institutional Review Board of The Hashemite University approved the study protocol (#2/7/2019/2020). Confidentiality was ensured by avoiding identifying information. Women were first provided with study information, then confirmed their understanding that participation was voluntary and they could stop answering the questions at any time if they wished to do so. They then gave consent online (at the start), confirmed by implication (at the end) through the return of completed questionnaires. Because of the anonymity, no written consent was obtained; instead, returning the completed questionnaire was taken as implied consent to participate. Contact details of the researchers were presented in the first page, giving support on request, to all participants. At the end of the EPDS items, a message was included saying that if you answered "Applies" to the last question (likely to harm oneself), you should see a doctor as soon as possible, or contact the researchers on their phone. We added a note on the cultural aspect of Muslim women, as belief in destiny helps them to accept difficult situations as being the will of Allah [40]. Data files were anonymous and kept securely in a personal computer with a password. Data are archived in Ulster University Institutional Repository.

## Results

The sample of 546 pregnant women ranged in age from 19 to 47 years (M = 28.7, SD = 4.7). Almost all (n = 543, 99.5%) were living with their husband within a nuclear family unit: with husband and own children (n = 466, 85.3%). Most (n = 435, 79.7%) were city dwellers. Whilst many women (n = 476, 87.1%) were educated to bachelor degree level, just over one third (37.4%, n = 204) were working. A few women smoked (n = 58, 10.6%), averaging less than one cigarette (M = 0.9) per day. On average, women slept 7.8 hrs (SD = 1.6, range = 4–16). The husbands' mean age was 33.4 years (SD = 5.7). Most of the husbands were working (n = 514, 94.1%) and many of them (n = 327, 59.9%) had a bachelor's degree or higher. The mean household income was 711.7 Jordanian dinar (JD) per month, varying widely (Equivalent to 1003.8 USD, SD = 1081.1).

The mean gestational age was 6.1 months (SD = 2.65), the mean gravida was 2.2 (SD = 1.4), the mean number of children was 1 (SD = 1.2), the mean number of miscarriages was 1.5 (SD = 0.9, 25.8%). The number of antenatal visits ranged from 0 to 6, ceasing altogether during quarantine. Most of the women had participated monthly antenatal visits (n = 449, 82.2%), where the average number of antenatal visits before the pandemic was 5.13 (SD = 3.1), but most of them (n = 504, 92.3%) were finding it difficult to continue with their antenatal visits during the pandemic. The majority of the women (n = 364, 66.7%) were visiting private clinics for antenatal care. Around one quarter (23.3%, n = 127) had received information about COVID-19 from their healthcare provider, and 70.9% (n = 90) of them considered that the level of information was sufficient. The women reported that their doctors (n = 59, 46.5%) and the internet including social media (n = 39, 30.7%) were their main sources of information about pregnancy and COVID-19.

As regards the presence of antenatal depression symptoms among the participants, the mean EPDS score was 11.4 (SD = 5.9, 95% CI was 10.9 to 11.9). The estimated prevalence of

pregnant women who had depressive symptoms was 36.8% (n = 201, 95% CI was 32.8% to 41%).

As for the KAP toward COVID-19 among the women, the knowledge mean score was 26.6 (SD = 3.4, 72% correct answers), the attitude mean score was 7.8 (SD = 0.53, 97.5 positive attitude), and the practice mean score was 10.9 (SD = 1.4, 84% correct practices). The total knowledge, attitude, and practice scores were dichotomized into high and low, based on the median.

Our data show the association of KAP and socio-demographical characteristics with antenatal depression symptoms (see S1 Table). First, an independent-samples t-test was conducted to compare the antenatal depression scores for the high and low knowledge scores. The result revealed that there was a significant difference in EPDS scores between the high knowledge score group (M = 10.8, SD = 5.7) and low knowledge score group [$M$ = 12.2, $SD$ = 6.1; t (544) = 2.7, $p$ = .007]. The magnitude of the difference in the means was large (eta squared = 0.13). Second, an independent-samples t-test was conducted to compare the antenatal depression scores for the low and high attitude score groups. The result showed that there was a non-significant difference in EPDS scores between the high attitude score group (M = 11.3, SD = 5.8) and low attitude score group [$M$ = 12.2, $SD$ = 6.4; t (544) = 1.3, $p$ = 0.2]. Third, an independent-samples t-test was conducted to compare the antenatal depression scores for the low and high practice score groups. The result indicated that there was a non-significant difference in EPDS scores between the high practice score group (M = 11.4, SD = 5.9) and low practice score group [$M$ = 11.5, $SD$ = 5.9; t (544) = 0.12, $p$ = 0.9].

Paired-samples t-tests were conducted to evaluate the impact of the socio-demographic characteristics and obstetrical variables on depression, showing that the EPDS was significantly ($p \leq 0.05$) lower among women who slept $\geq 8$ hours, who had a higher household income ($\geq 500$ JD [705 USD]), who had a higher level of education (diploma or higher), who lived in a city, who were employed, earlier in pregnancy (less than six months of gestation), whose husband was employed, and whose husband had a higher level of education. Neither age (*woman or husband*), type of family unit, number of antenatal visits, gravida, nor number of live children had any statistically significant assocaition with the EPDS score.

## Discussion

This is the first study undertaken with pregnant women during the COVID-19 quarantine in the Kingdom of Jordan. It is also the first study to examine the association of KAP toward COVID-19 with antenatal depressive symptoms among pregnant women.

When we compared the socio-demographical characteristics of the participants in the current study with those in the most recent Jordan Population and Family Health Survey (JPFHS) [41], we found that the participants were representative of the ever-married women in the JPFHS who use the internet almost every day rather than general ever-married women in Jordan. The age of the women in the current study ranged from 19–47 yrs: the women in the JPFHS were aged 15 to 49 yrs. Most of the participants in the current study were living in the city (79.7%) and most were educated to diploma level or higher (87.1%). Likewise, in the JPFHS, most of the women lived in an urban area (85.2%) and most had completed a diploma or higher level of education (95.3%).

We found that the prevalence of depressive symptoms among Jordanian pregnant women during quarantine was 36.8%. This finding is comparable to the regional and international studies during the pandemic, who used EPDS. For example, 44.2% of Egyptian pregnant women [24] and 45.9% of Chinese pregnant women were having depressive symptoms during the COVID-19 outbreak [25]. However, research in Greece reports that antenatal depression remained at 13.5% during the pandemic [26].

 

Due to the nature of recruitment and data collection of this study (online), where the population responding to the survey may be very different from the population included in the historical study, it is not ideal to compare our current findings with previous ones. We can see that our finding is almost double that reported in 2011 when the equivalent figure was 19% [39]. However, it is lower than the figure reported in our 2014 study [6], where 57% of women reported depressive symptoms during pregnancy. Also, due to the nature of the data collection, we could not infere that our results support the supposition that depressive symptoms during the COVID-19 quarantine would increase among pregnant women or not like other studies, where they noted increase depressive symptoms among healthcare providers [31, 32]. We need to keep in mind that pregnant women in Jordan have been asked to stay home during the pandemic and were not in the frontline in the battle against COVID-19 like healthcare providers, and this could help them to feel safe and improve their psychological status.

Our findings should be interpreted with caution as they could be influenced by the sampling and survey techniques used (snowballing and *Google Forms*): the topic needs to be monitored after the pandemic has receded. It is helpful to note that recent other studies reported different percentages of depressive symptoms: China 45.9% [25], Greece 13.5% [26], Egypt 44.2% [24].

We saw lower EPDS scores were associated with better knowledge: 13% of the variance *(i.e. a relatively large effect)* in depressive symptoms was explained by COVID-19 knowledge. Women who have higher levels of knowledge have fewer symptoms of antenatal depression. One explanation for this finding is that when women have better knowledge and up-to-date information about COVID-19 they feel more prepared and relaxed which then results in them experiencing fewer psychological problems. As the effect size was large, it suggests that if pregnant women knowledge about COVID improved, they may experience better mental health. This suggestion is supported by research in Wang, Pan et. al [28], where provision of information about COVID-19 reduced depression levels and lowered the psychological impact of the outbreak among the general population.

Almost all (97.5%) of the participants had a positive attitude about COVID-19 and most of them (84%) performed correct practices to combat the disease. However, the EPDS scores were not significantly affected by attitude and practice. This means that positive attitude and correct practice were not seen to affect the antenatal depression level of pregnant women. This contradicts earlier report [28] that people who performed correct practices for COVID-19 prevention, such as hand-washing, experienced less antenatal depression. This could be explained by the fact that the pregnant women in Jordan are Muslims and they wash their hands five times per day as a part of their ablutions prior to prayer.

As regards socio-demographics, we found that there were statistically significant lower EPDS scores among women who slept longer, had a higher household income, had a higher level of education, lived in a city, were employed, had a gestation of less than six months, had a husband who was employed with a higher education level. This suggests that the pregnant women were at lower risk for developing depressive symptoms if they were from a high socio-economic class, had adequate rest, and were in their first trimester. The negative association between socioeconomic class and depression has been seen before Silva, Jansen [42]. Similarly, our finding that women who slept more had fewer depressive symptoms than others is congruent with Okun, Kiewra [43] who used the Hamilton scale for depression and found that sleep is more disturbed in depressed pregnant women. Moreover, the result in regarding women in early pregnancy having lower depressive symptoms than women in late pregnancy is congruent with Howdeshell and Ornoy [2], who found that antenatal depression is higher in the later trimesters and increases as pregnancy progresses (7.4% in the first trimester and 12.0%–12.8% in the second and third trimester).

 

Finally, we found that age, type of family unit, husband's age, number of antenatal visits, gravida, and family size (number of children) was not significantly associated with depressive symptoms. The non-significant finding for age was reported previously [6].

## Limitations

First, because the questionnaire was distributed online, pregnant women without access to the internet were unable to take part in the survey, which may have resulted in an under-representative sample in favour of those more highly educated and employed. Hence, our findings are not fully generalisable to the whole population of pregnant women in Jordan Second, some of the KAP measures were newly developed by authors and due to time limitations, were validated by a limited number of experts, whereas ideally, multidimensional measures should be developed via focus group discussion and in-depth interview [27]. Third, we note the inability to make definite causal inferences due to the cross-sectional design. Finally, some aspects of the CHERRIES checklist [36] were not feasible, including: randomisation of items (due to the content and logical flow of the questions), ensuring unique site visitors, view rate, participation rate, completion rate, cookies used, IP check, and Log file analysis (as it was an anonymous free survey).

## Implications

The findings of this study have implications for practice, policymakers, education, and further research. In respect of practice, plans for the screening of antenatal depression should be considered for emergency situations, such as the COVID-19 pandemic. Also, midwives and nurses need to find innovative ways to reach pregnant women and educate them about current health risks, such as COVID-19, in order to protect the mental health of pregnant women. One possibility could be found through online antenatal education that includes mental health promotion. Policymakers should disseminate specific educational messages on prevention targeted at pregnant women (for example through radio, television, and social media channels). This would be likely to increase their adherence to the control measures and lower their levels of antenatal depression. Furthermore, policymakers should also provide healthcare professionals with continuing education about current health risks to enable the knowledge to be cascaded to families and so improve their knowledge, attitudes and practice. We suggest that university curricula should include appropriate and effective responses to emergencies such as the COVID-19 pandemic in order to increase students' knowledge, which (in addition to more direct clinical benefits) would result in improving their ability to protect the mental health of pregnant women [33]. Further research on pregnant women from all socioeconomic backgrounds is needed in order to ascertain whether there are differences in depressive symptoms and KAP across the wider socioeconomic spectrum.

## Conclusion

Both healthcare providers and the wider community have a responsibility to protect pregnant women during emergency situations such as the COVID-19 pandemic. Pregnancy is a critical period for mother and baby and every possible care should be taken to ensure the best possible outcomes. We found that depressive symptoms were prevalent among Jordanian pregnant women and that these symptoms were significantly associated with lower levels of knowledge about COVID-19, but not with attitude and practice toward COVID-19. We found that women from a higher socioeconomic class had lower levels of depressive symptoms as compared to women from a lower socioeconomic class.

## Supporting information

**S1 File. Survey announcement.** A survey of the knowledge, attitudes and practices of pregnant Jordanian women towards the coronavirus disease (COVID-19) during the outbreak period in Jordan.
(PDF)

**S2 File. Knowledge, attitudes, and practices of Jordanian pregnant women towards coronavirus disease (COVID-19) during the period of its outbreak: Cross-sectional survey (Arab version).**
(DOCX)

**S3 File. Knowledge, attitudes, and practices of Jordanian pregnant women towards coronavirus disease (COVID-19) during the period of its outbreak: Cross-sectional survey (English version).**
(DOCX)

**S1 Checklist. Checklist for reporting results of internet E-surveys.**
(DOCX)

**S1 Table. Knowledge, attitudes, practice and socio-demographic characteristics with symptoms of depression among Jordanian pregnant women, showing significant associations, e.g. higher knowledge level was associated with lower depression.**
(DOCX)

## Acknowledgments

We would like to acknowledge all the Jordanian pregnant women who participated in our study.

## Author Contributions

**Conceptualization:** Sanaa Abujilban, Lina Mrayan, Salwa Al-Obeisat.

**Data curation:** Sanaa Abujilban, Lina Mrayan, Salwa Al-Obeisat, Mu'ath Tanash, W. George Kernohan.

**Formal analysis:** Sanaa Abujilban, Mu'ath Tanash.

**Investigation:** Sanaa Abujilban, Mu'ath Tanash.

**Methodology:** Sanaa Abujilban, Lina Mrayan, Mu'ath Tanash, W. George Kernohan.

**Project administration:** Sanaa Abujilban, Mu'ath Tanash.

**Resources:** Sanaa Abujilban, Lina Mrayan, Mu'ath Tanash, W. George Kernohan.

**Software:** Mu'ath Tanash.

**Supervision:** Sanaa Abujilban, Mu'ath Tanash, Marlene Sinclair, W. George Kernohan.

**Validation:** Salwa Al-Obeisat, Mu'ath Tanash, Marlene Sinclair.

**Writing – original draft:** Sanaa Abujilban, Lina Mrayan, Salwa Al-Obeisat.

**Writing – review & editing:** Sanaa Abujilban, Lina Mrayan, Salwa Al-Obeisat, Marlene Sinclair, W. George Kernohan.

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
