## [Decision Letter · Decision Letter 0]

23 Sep 2021

 PGPH-D-21-00532

 Factors influencing levels of antenatal depression

in the Kingdom of Jordan during the COVID-19 pandemic

Dear Dr. Kernohan,

Thank you for submitting your manuscript to PLOS Global Public Health. After careful consideration, we feel that it has merit but does not fully meet PLOS Global Public Health’s publication criteria as it currently stands. Therefore, we invite you to submit a revised version of the manuscript that addresses the points raised during the review process.

 As you will have noticed, both reviewers recommend major revisions to your article. It would be most helpful if you could focus on the detailed points provided by Reviewer #1, and, in particular, pay special attention to the limitations of online surveys as you revise.

We look forward to receiving your revised manuscript.

Kind regards,

Khameer Kidia

Academic Editor

Reviewers' comments:

Reviewer's Responses to Questions

Journal Requirements:

1. Please include additional information regarding the survey or questionnaire used in the study and ensure that you have provided sufficient details that others could replicate the analyses. For instance, if you developed a questionnaire as part of this study and it is not under a copyright more restrictive than CC-BY, please include a copy, in both the original language and English, as Supporting Information.

2. Please update the completed 'Competing Interests' statement, including any COIs declared by your co-authors. If you have no competing interests to declare, please state "The authors have declared that no competing interests exist". Otherwise please declare all competing interests beginning with the statement "I have read the journal's policy and the authors of this manuscript have the following competing interests:"

**Comments to the Author**

1. Does this manuscript meet PLOS Global Public Health’s publication criteria? Is the manuscript technically sound, and do the data support the conclusions? The manuscript must describe methodologically and ethically rigorous research with conclusions that are appropriately drawn based on the data presented.

Reviewer #1: Partly

Reviewer #2: Partly

2. Has the statistical analysis been performed appropriately and rigorously?

Reviewer #1: No

Reviewer #2: No

3. Have the authors made all data underlying the findings in their manuscript fully available (please refer to the Data Availability Statement at the start of the manuscript PDF file)?

Reviewer #1: Yes

Reviewer #2: Yes

4. Is the manuscript presented in an intelligible fashion and written in standard English?

Reviewer #1: Yes

Reviewer #2: Yes

5. Review Comments to the Author

Reviewer #1: General Comment:

The authors present original research covering a highly topical condition with grave long-term consequences for pregnant women and their families. They rightly note that monitoring mental wellbeing should be prioritised in the current era. The manuscript is well-written and accessible. While the findings are very interesting, the paper would benefit from editing to refine its focus and shorten its length. There are challenges with the inferences made from the data obtained through this study design that require attention to avoid understating the limitations and overstating the implications. However, the paper shows sufficient potential for publication and the authors are encouraged to resubmit a revised version.

Major revisions:

Introduction.

1. This section is too long and unfocussed. For example, consider removing the text about the risk of COVID-19 disease in pregnancy (lines 84-100) and confine the content of the Introduction to psychological aspects of pregnancy in the COVID-19 era and the restrictive measures instituted during national lockdowns.

2. Line 126…and to examine the effect of KAP toward COVID-19 and socio-demographics on antenatal depression symptoms among Jordanian pregnant women. A cross-sectional, correlational study design was employed. The inherent nature of this research method limits the conclusions that can be inferred about causality; a cross-sectional design can identify associations but can’t examine the ‘effect’ of a variable on the outcome. Please review the stated aim with this in mind and consider rewording to align with the accepted limitation stated in line 327 in the Discussion (Finally, we note the inability to make definite causal inferences due to the cross-sectional design.)

3. Lines 135-136, research questions. As above, the study design selected can’t answer the questions posed: Did depressive symptoms in Jordanian pregnant women increase during quarantine in comparison to the reported previous levels? Using an historical comparison is not ideal, primarily because the recruitment and data collection approaches were vastly different (in-person vs online) and online surveys are often not generalisable. For example, pregnant women with moderate-severe depression could be less willing than women with none to engage in an online survey, or could begin but not complete/submit their responses. Conversely, pregnant women with depressive symptoms may be more inclined to be highly interested to engage with online content about mental health or be singled out by their friends during snowballing and encouraged to complete the survey. The population responding to the survey may be very different from the population included in the historical study, so the proportion of women with depressive symptoms observed in this study applies exclusively to the sample of women who completed this survey. Please review all statements made about the hypothesis that depression would ‘increase’ during the lockdown with this in mind (including lines 273-277 in the Discussion).

4. Lines 136-137, research questions. Do KAP toward COVID-19 and socio-demographics have an effect on antenatal depression symptoms? As above, the cross-sectional design is able to identify sociodemographic variables associated with depression symptoms but not examine whether they ‘have an effect’.

Methods and Materials.

5. Consider completing the CHERRIES checklist to ensure all aspects of reporting for online surveys are covered: Eysenbach, G. (2004). Improving the quality of web surveys: the checklist for reporting results of internet e‐surveys (cherries). Journal of medical Internet research, 6(3)e34 doi:10.2196/jmir.6.3.e34 http://www.jmir.org/2004/3/e34/ The categories requiring additional detail include survey administration, response rates and preventing multiple entries from the same individual. Aspects of the checklist that were not feasible within your study could be mentioned as Limitations.

6. Line 187. Analysis. The primary outcome is estimated prevalence of depressive symptoms reported as a percentage. Calculating confidence intervals around that point value would provide more information about its precision. Please consider doing so.

7. Line 195. Ethical considerations. This section should cover potential risks and benefits of study participation, as well as ethical oversight, informed consent and confidentiality. Due to the anonymity of the online survey, respondents with depressive symptoms could not be traced to provide care and/or treatment, presumably. Please mention any efforts made by the researchers to enquire if pregnant women engaged with the online survey needed to talk with trained counsellors or to link respondents with available resources to promote wellbeing.

Discussion.

8. Lines 278-282. The cautionary note in the Discussion about generalisability is well stated. The paper would be strengthened by concentrating on the prevalence observed and the associated socio-demographic and medical factors identified, rather than making comparisons with historical studies of antenatal depression and other populations (frontline health workers) and providing explanations that are pure speculation (line 275-277, 298-300).

Minor revisions:

9. Line 164. In the current study, the EPDS was found to have a Cronbach’s alpha of 0.85, indicating high reliability. This sentence is confusing because the study being submitted for publication does not involve validation procedures. Please clarify.

10. Line 377. This would be likely to increase their adherence to the control measures and lower their levels of antenatal depression. This effect of increasing education and awareness can’t be implied by the data presented; it is possible (vs would be likely) but would need to be assessed.

Reviewer #2: The article describes an important area area. However, the authors have not described their methodology fully for the reader to understand how they collected their data. It is unclear if the participants had covid infection or they were just pregnant women.

The use of the term "quarantine" is so unclear here. Do the authors refer to quarantine as those lockdown measures instituted to prevent the spread of covid infection; not necessarily separating a person with covid infection to spread its infection.

The authors should ensure that they reference their work. They should also revise the way their refence list appears.

See attached document

6. PLOS authors have the option to publish the peer review history of their article (what does this mean?). If published, this will include your full peer review and any attached files.

**Do you want your identity to be public for this peer review?** For information about this choice, including consent withdrawal, please see our Privacy Policy.

Reviewer #1: No

Reviewer #2: **Yes: **Malinda Kaiyo-Utete

---

## [Decision Letter · Decision Letter 1]

19 Jan 2022

Factors associated with antenatal depression

in the Kingdom of Jordan during the COVID-19 pandemic

PGPH-D-21-00532R1

Dear Dr. Kernohan,

We're pleased to inform you that your manuscript has been judged scientifically suitable for publication and will be formally accepted for publication once it meets all outstanding technical requirements.

Within one week, you'll receive an e-mail detailing the required amendments. When these have been addressed, you'll receive a formal acceptance letter and your manuscript will be scheduled for publication.

An invoice for payment will follow shortly after the formal acceptance. To ensure an efficient process, please log into Editorial Manager at https://www.editorialmanager.com/pgph/ click the 'Update My Information' link at the top of the page, and double check that your user information is up-to-date. If you have any billing related questions, please contact our Author Billing department directly at authorbilling@plos.org.

Kind regards,

Khameer Kidia

Academic Editor

Additional Editor Comments (optional):

Reviewers' comments:

Reviewer's Responses to Questions

**Comments to the Author**

1. If the authors have adequately addressed your comments raised in a previous round of review and you feel that this manuscript is now acceptable for publication, you may indicate that here to bypass the “Comments to the Author” section, enter your conflict of interest statement in the “Confidential to Editor” section, and submit your "Accept" recommendation.

Reviewer #1: All comments have been addressed

Reviewer #2: All comments have been addressed

2. Does this manuscript meet PLOS Global Public Health’s publication criteria? Is the manuscript technically sound, and do the data support the conclusions? The manuscript must describe methodologically and ethically rigorous research with conclusions that are appropriately drawn based on the data presented.

Reviewer #1: Yes

Reviewer #2: Partly

3. Has the statistical analysis been performed appropriately and rigorously?

Reviewer #1: Yes

Reviewer #2: Yes

4. Have the authors made all data underlying the findings in their manuscript fully available (please refer to the Data Availability Statement at the start of the manuscript PDF file)?

Reviewer #1: Yes

Reviewer #2: Yes

5. Is the manuscript presented in an intelligible fashion and written in standard English?

Reviewer #1: Yes

Reviewer #2: Yes

6. Review Comments to the Author

Reviewer #1: (No Response)

Reviewer #2: The authors have addressed the previous comments

7. PLOS authors have the option to publish the peer review history of their article (what does this mean?). If published, this will include your full peer review and any attached files.

**Do you want your identity to be public for this peer review?** For information about this choice, including consent withdrawal, please see our Privacy Policy.

Reviewer #1: No

Reviewer #2: No
